# Pelagic *Sargassum* as a Potential Vector for Microplastics into Coastal Ecosystems

Dalila Aldana Arana [1], Tania P. Gil Cortés [1], Víctor Castillo Escalante [1] and Rosa E. Rodríguez-Martínez [2,*]

1 CINVESTAV IPN Unidad Mérida, Km. 6 Antigua Carretera a Progreso Cordemex, Merida 97310, Yucatan, Mexico; daldana@cinvestav.mx (D.A.A.); tania.gil@cinvestav.mx (T.P.G.C.); vicas@cinvestav.mx (V.C.E.)

2 Unidad Académica de Sistemas Arrecifales-Puerto Morelos, Instituto de Ciencias del Mar y Limnología, Universidad Nacional Autónoma de México, Puerto Morelos 77580, Quintana Roo, Mexico

* Correspondence: rosaer@cmarl.unam.mx

**Abstract:** Macroalgal blooms are increasing globally, with those linked to pelagic *Sargassum* affecting over 30 nations since 2011. As *Sargassum* mats traverse the Atlantic Ocean and the Caribbean Sea, they entrap and transport plastic to coastal areas, intensifying pollution in diverse ecosystems. This research assessed microplastics (MPs) within *Sargassum fluitans* III collected from the northern Mexican Caribbean coast (March 2021 to January 2022). The study employed a hydrogen peroxide protocol for macroalgae pretreatment to optimize MP extraction. All samples analyzed contained MPs at monthly mean concentrations that ranged from 3.5 to 15.3 MPs g$^{-1}$ DW, with fibers constituting ≥90%. Fiber colors, mainly transparent, blue, and black, exhibited diverse sizes and wear stages. The study underscores the pervasive and consistent presence of MPs in pelagic *Sargassum* reaching the Mexican Caribbean. Considering the documented *Sargassum* influxes to this coast in recent years (2789–11,297 tons km$^{-1}$ yr$^{-1}$), potential annual MP influxes range from $0.1 \times 10^9$ to $17.3 \times 10^9$ km$^{-1}$ yr$^{-1}$. Efficiently removing beach-cast *Sargassum* and directing it to landfills could serve as a viable strategy for the simultaneous removal of attached MPs from the ocean and coastal waters, offering a promising mitigation strategy to combat plastic pollution in the examined marine environment.

**Keywords:** macroalgal bloom; coastal pollution; microplastics; Caribbean; Mexico

## 1. Introduction

Since the 1970s, macroalgal blooms have significantly increased, impacting biodiversity, human health, and economies [1,2]. Noteworthy instances include green tides of *Ulva* spp. [2,3], *Cladophora* sp., and *Enteromorpha* [4], and brown tides of *Sargassum horneri* [5,6]. Since 2011, two holopelagic *Sargassum* species (*S. fluitans* morphotype III and *S. natans* morphotypes I and VIII) have substantially increased in abundance in the tropical Atlantic, impacting over 30 nations due to the recurrent formation of a Great Atlantic *Sargassum* Belt, extending from West Africa to the Gulf of Mexico [7].

*Sargassum* decay along coastlines causes severe ecological and socio-economic impacts. The leachates produced deteriorate the quality of coastal waters and cause the mortality of flora and fauna [8,9], and the gases (e.g., hydrogen sulfide, ammonia, and methane) contribute to climate change and affect human health [10–12]. Removing *Sargassum* from beaches and coastal waters is a costly endeavor. Many affected countries lack the resources for widespread cleanup efforts, often leaving most affected areas unattended.

While exploring the potential uses of *Sargassum* for food, biofuels, construction materials, or pharmaceutical products could help offset removal and management expenses [13,14], thoroughly investigating its elemental composition and potential hazards is crucial for safe utilization. While these algae offer a rich array of bioactive compounds, minerals, trace elements, amino acids, and fatty acids that can be harnessed for agricultural

purposes and animal feeding [15], they also possess the capacity to absorb potentially toxic elements, including arsenic [16,17] and chlordecone [18], which limit their usage in large quantities.

As *Sargassum* mats traverse the Atlantic Ocean, they also trap plastic waste and transport it to shorelines, constituting an additional source of pollution [19]. Plastic pollution has become the primary form of anthropogenic debris in marine environments, accounting for up to 80% of marine litter [20]. It has been estimated that 12.7 million metric tons of plastic find their way into the oceans annually [21,22]. In the sea, macroplastics (5–50 cm) can break down into microplastics (≤5 mm) and nanoplastics (≤0.1 μm) when exposed to UV radiation and subjected to mechanical wear from waves. Microplastics can also reach the ocean through continental discharges. According to estimates by [23], between 82 and 358 trillion plastic particles are floating in the sea, constituting 1.1 to 4.9 million tons of pollution.

Microplastics (MPs) pervade ecosystems spanning from polar regions to tropical areas. Characteristics such as color, shape, morphology, and degradation stage offer crucial insights into their origin, age, and weathering [24]. Their morphology significantly influences transport and fate in diverse environments [25]. Moreover, both morphology and color play pivotal roles in influencing the ingestion of MPs by marine organisms [26].

These minute plastic particles and fibers have been found in virtually all marine species [27], bioaccumulating in tissues and traversing the food chain, ultimately reaching humans through seafood consumption [28]. Microplastics also act as surfaces for hydrophobic organic contaminants, like pesticides and polychlorinated biphenyls [29,30], and can serve as habitats for various bacterial and eukaryotic organisms [31], some of which may be potentially pathogenic [32,33]. The presence of MPs in beach-cast *Sargassum* poses a potential threat to multiple ecosystems and may constrain its valorization.

The Caribbean Sea ranks as the second most plastics-polluted sea globally [34], yet there is a notable absence of specific regulations governing marine litter or MPs in this region. Various marine environments in the Caribbean, including Brazil, Colombia, Jamaica, Antigua, Bonaire, Aruba, and San Blas, have reported the prevalence of polyethylene and polypropylene microplastics [35–38]. The presence of MPs in beach sediments was documented in Colombia, Brazil, Mexico, and Puerto Rico [39–42]. Microplastics have infiltrated diverse ecosystems, such as seagrass beds of *Thalassia testudinum*, where 75% of the leaves may contain MPs, primarily fibers [43]. The Queen conch (*Aliger gigas*), a bioindicator of microplastic pollution in marine organisms, exhibits alarming contamination levels ranging from 43 to 270 MPs per individual, with fibers being the predominant type [44].

This study aims to evaluate the abundance of microplastics and their monthly variability within pelagic *Sargassum* specimens collected from the shoreline along the northern Mexican Caribbean coast. The results will contribute to estimating the potential annual influx of microplastics to the coastline, drawing on historical landing volumes of *Sargassum*.

## 2. Materials and Methods

### 2.1. Study Area

The study was conducted in Puerto Morelos, located on the northeastern coast of the Yucatan Peninsula (Figure 1A). Massive landings of *Sargassum* in this area commenced in late 2014 [45] and have been an annual occurrence since 2018, with high-volume stranding typically lasting from five to seven months, particularly from March to September [46]. The quantity of beach-cast *Sargassum* during peak landing months along this coast increased from an average of 2360 cubic meters per kilometer in 2015 [45] to 6565 cubic meters per kilometer in 2019 [46].

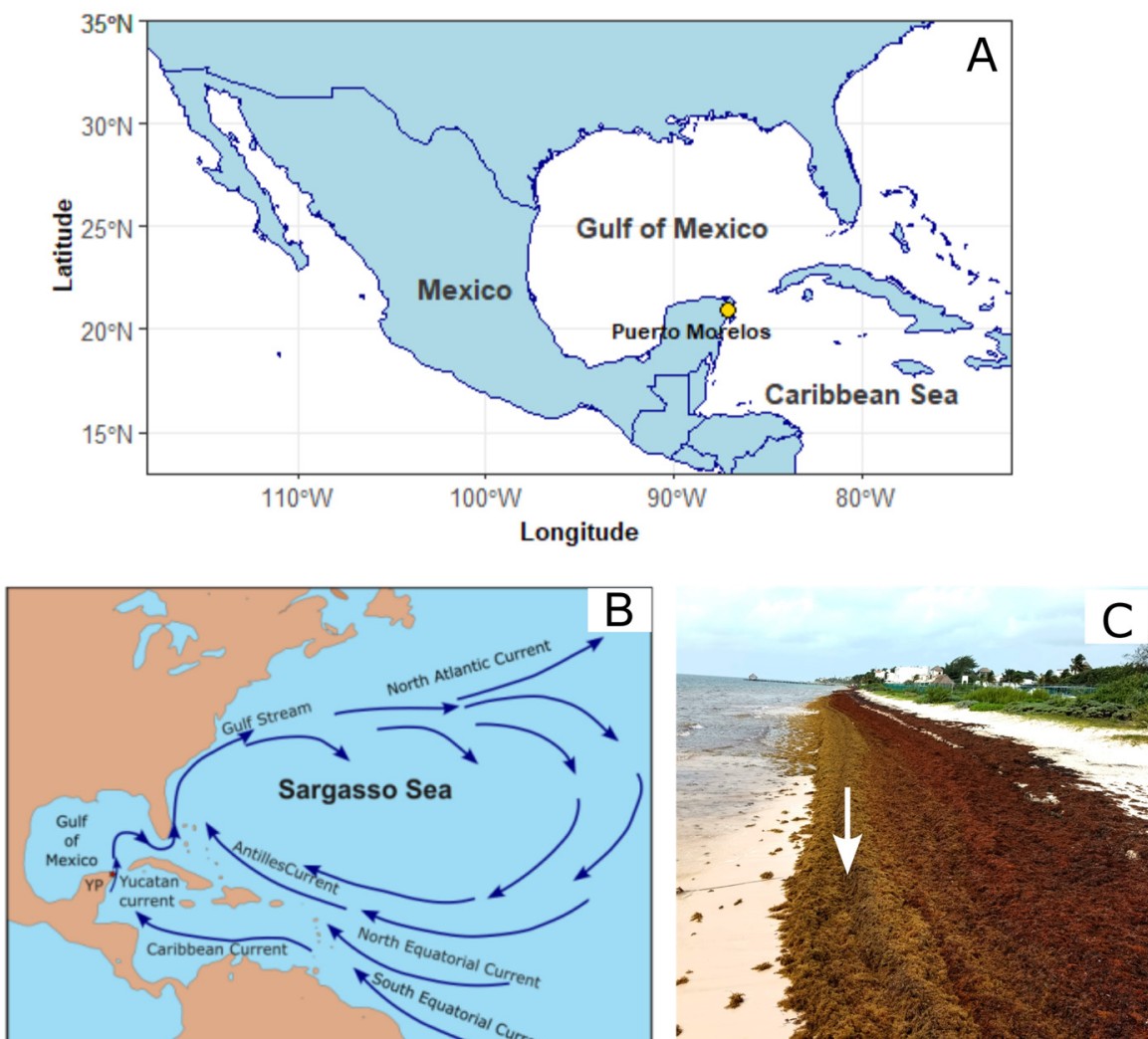

**Figure 1.** (**A**) *Sargassum* sampling location. (**B**) Schematic view of the significant currents in the North Atlantic. (**C**) *Sargassum* beach inundation. The arrow indicates recently deposited *Sargassum* (golden in color), which was used for measuring the abundance of microplastics.

The *Sargassum* masses observed along this coastline during the study period are likely to have originated from the Great Atlantic *Sargassum* Belt [7] and were transported to the Mexican Caribbean coast by the South Equatorial Current, the Caribbean Current, and the Yucatan Current (Figure 1B). The wind regime significantly influences the quantity of *Sargassum* landing; east and southeast winds facilitate its movement towards the Mexican coast during spring and summer, while northerly winds inhibit onshore transport during autumn and winter [47]. Data on sea surface temperature and prevailing sea wind direction and speed obtained daily at 1 min intervals for the study period were provided by the Oceanographic and Meteorology Monitoring Service of UNAM in Puerto Morelos (SAMMO), which has an ISO 9001 certification [48]. Wind speeds during the study ranged from 2.4 to 15.3 m s$^{-1}$, with monthly means of 4.5 and 7.4 m s$^{-1}$, originating primarily from the east to the southeast (Figure 2). The monthly sea surface temperature ranged from 26.0 to 30.6 °C.

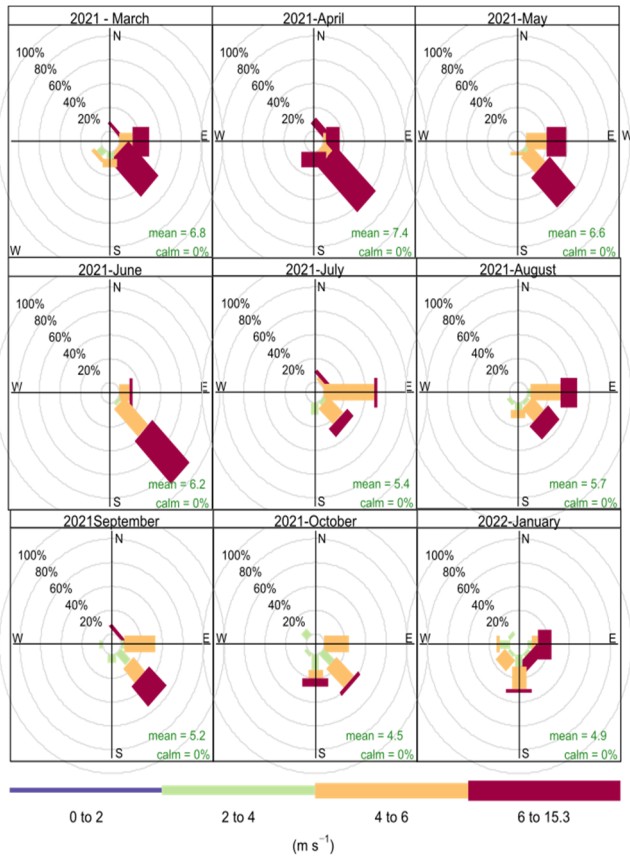

**Figure 2.** Windrose plots of mean wind speed and direction frequencies in March–October 2021 and January 2022 in the northeastern Mexican Caribbean. Wind speeds are split into the intervals shown by the scale at the bottom of both panels. The gray circles show the percent frequencies.

## 2.2. Sample Collection and Treatment

Fresh *Sargassum fluitans* III was meticulously collected by hand from specimens recently washed ashore in proximity to the Unidad Académica de Sistemas Arrecifales, Instituto de Ciencias del Mar y Limnología, UNAM (Figure 1C). The collected samples were promptly placed into a previously washed plastic box and sun-dried. Upon complete drying, the entire box contents were carefully transferred to a Ziploc bag to prevent the loss of any particulate matter that might have dislodged during the process. Species identification followed the methods outlined in [49]. Monthly collections were conducted from March to October 2021 and in January 2022. Three samples were collected each month. The average wet weight of the samples was 99.4 g (SE: 25.6). After sun drying, the mean weight was reduced to 22.4 g (SE: 10.2). Subsequently, the samples were dispatched to CINVESTAV IPN Unidad Mérida.

Five grams of *Sargassum* were extracted from each sample in the laboratory. These samples were placed in individual beakers containing 50 mL of 30% hydrogen peroxide, ensuring the frond was fully submerged. The treatment underwent a 48 h digestion period at a room temperature of 22 °C to remove a portion of the organic matter and induce discoloration in the *Sargassum* to facilitate visualization of MPs through a stereoscope; this methodology was adapted from [50]. Each beaker was covered by aluminum foil during digestion to prevent contamination by microplastics from the environment.

Afterward, the liquid was transferred into a test tube, and 40 mL of distilled water was added to the beaker with the macroalgae to rinse out. This process was repeated thrice, with manual agitation for each rinse for one minute. The *Sargassum* thalli were then extracted using dissecting forceps and placed in a Petri dish for examination to check

for any remaining adhering microplastics. Microplastics were also counted on the liquid obtained from the rinses.

### 2.3. Optical Identification

Individual *Sargassum* thalli were carefully positioned in a Petri dish, allowing for examination of both sides with dissecting forceps. The abundance and diversity of microplastics in both the liquid and *Sargassum* thalli were assessed using a Leica Zoom stereoscopic microscope (Motic SMZ-171 Series, Motic, Kowloon, Hong Kong) at 40× magnification. All ocularly observable microfibers were tallied. The entire liquid sample was examined within a gridded Petri dish to prevent miscounts. Identified MPs were categorized by form (fiber, fragment, film, sphere) and color [28]. The number of MPs within the 500 to 5000 µm size range was counted. To avoid miscounts, the entire surface area of each filter was examined using gridlines, and the MPs on each filter were counted twice to ensure accuracy. Some microfibers (*n* = 84) were precisely measured in electron microscopy using its automatic measurement program or the scale in the microphotographs. Their relative frequency was determined in five size ranges, with one-thousand-micron intervals.

### 2.4. Scanning Electron Microscopy (SEM) and Energy-Dispersive X-ray Spectroscopy (EDXS)

After identifying and quantifying the microplastics, five were placed on a lead base and then metalized with a gold–palladium coating (Quorum Q150R ES, Quorum Technologies Ltd. Ashford. Kent. England) for 45 s. This prepared them for examination under a scanning electron microscope, specifically the model JEOL JSM-7600F (Jeol, Schottky, Japan).

### 2.5. Statistical Analysis

Due to violations of assumptions regarding normality and homoscedasticity, we estimated the 95% confidence intervals of the monthly mean using bootstrap resampling with 1000 replicates using Infosat software. Statistical comparisons among these confidence intervals were based on [51].

### 3. Results

#### 3.1. Microplastics Concentration

The monthly microplastics (MPs) in *Sargassum* per sample (5 g) ranged from 16 to 98, resulting in monthly means fluctuating from 3.5 to 15.3 MPs g$^{-1}$ DW (Figure 3). The peak mean MP concentration was recorded in April 2021, while the lowest was observed in June 2021. It is worth noting that the relative abundance of MPs that persisted on *Sargassum* after rinsing ranged from 24% to 72% of the monthly samples. This emphasizes that relying solely on rinsing the blades is insufficient for the complete removal of MPs.

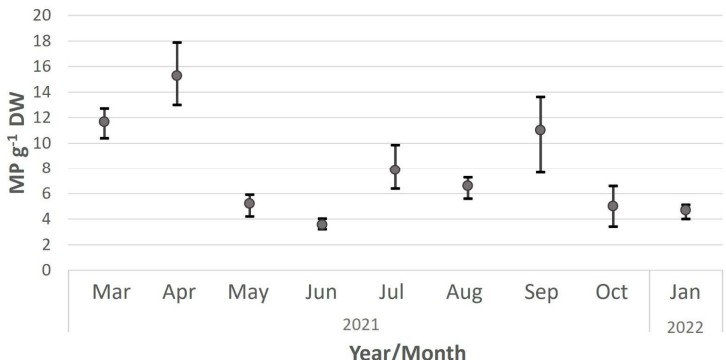

**Figure 3.** Fluctuation in *Sargassum*'s microplastic (MP) concentration gathered from the Puerto Morelos shoreline in México from March to October 2021 and January 2022. The circle denotes the mean, and the vertical lines indicate the 95% confidence intervals calculated through bootstrap resampling (N = 1000).

## 3.2. Microplastic Forms

Throughout the study period, fibers consistently constituted the predominant type of MPs in *Sargassum* (91%), followed by fragments (8.3%); films (0.6%) and spheres (0.1%) were rare. The dominance of fibers remained relatively stable across the sampling months, except for October 2021, when fragments comprised 52% of the total microplastics observed (Figure 4).

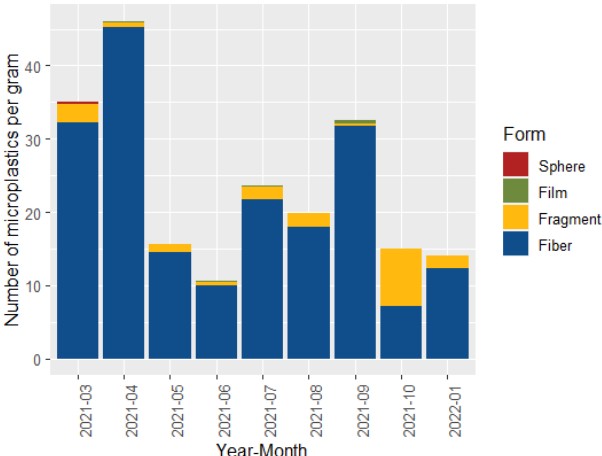

**Figure 4.** Monthly abundance of microplastics (N per gram) removed from *Sargassum* collected from Puerto Morelos, Mexico, from March 2021 to January 2022 in relation to their form.

## 3.3. Microplastic Colors

Most fibers identified were either transparent, black, or blue, with smaller proportions of red and multicolored (Figure 5). White, violet, gray, green, brown, pink, orange, gold, silver, and yellow fibers were infrequently observed and grouped under "Other". Fragments were primarily black and blue, with red and "Other" colors recorded occasionally (Figure 5). Some fragments broke during manipulation, underscoring their fragility, likely associated with wear and tear. The few films encountered were multicolor, blue, or "Other" colors; the only sphere recorded was black. Colored fibers are easy to locate in optical microscopy, as well as transparent fibers, which appear very bright in this type of microscopy. Since most of the fibers found were transparent, MP underestimation is not likely.

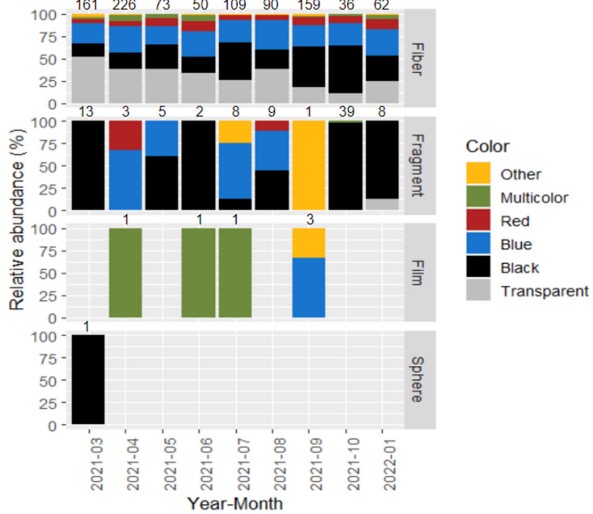

**Figure 5.** Relative abundance of colors observed in four types of microplastics (MPs) removed from *Sargassum* collected from Puerto Morelos, Mexico, from March 2021 to January 2022. The number above each bar indicates the monthly sample size for each type of MP.

### 3.4. Microplastic Size Distribution

Microplastic size was measured in 84 out of 1139 counted. Their length varied from 505 to 5073 μm, with a mean (SD) of 1935 μm (±1147). The majority fell in the size range accepted for MPs between 500 and 5000 μm. The most frequent size category was within the 1001–2000 μm range, representing 45% of all MPs, followed by the size category of 2001–3000 μm with 26%. Those remaining fell within the 500–99 μm (14%), 3001–4000 μm (6%), and 4001–5000 μm (9%) categories (Figure 6).

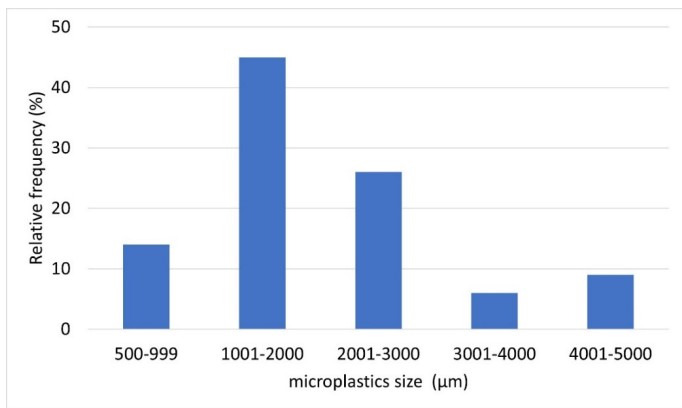

**Figure 6.** Size frequency distribution of microplastics attached to *Sargassum fluitans* III.

Figure 7 presents detailed MP photographs captured using a stereoscopic microscope. In Figure 7A, a black MP fragment is displayed. The images in Figure 7B–F depict fibers in diverse colors, sizes, and stages of wear, providing a comprehensive view of the variability in their characteristics.

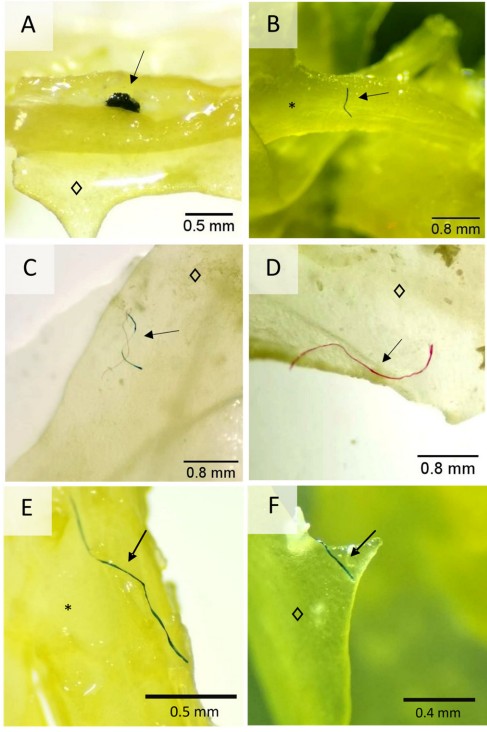

**Figure 7.** Stereoscopic microscope images of microplastics in *Sargassum* collected from Puerto Morelos, Mexico. The arrows on each figure point to: (**A**) black microfragment on a blade, (**B**) black microfiber on a stipe, (**C**) bicolor microfiber on a blade, (**D**) red microfiber on a blade, (**E**) blue microfiber on a stipe, and (**F**) blue microfiber on a blade. *: stipe, ◇: blade.

Images of microplastics captured through a scanning electron microscope (SEM) unveiled diverse manifestations of wear, transverse ruptures, and cavities, as illustrated in Figure 8A–C. These images also revealed objects resembling coccoid and bacillar bacteria (Figure 8E,F). However, additional studies are imperative to confirm the presence of microorganisms in these contexts.

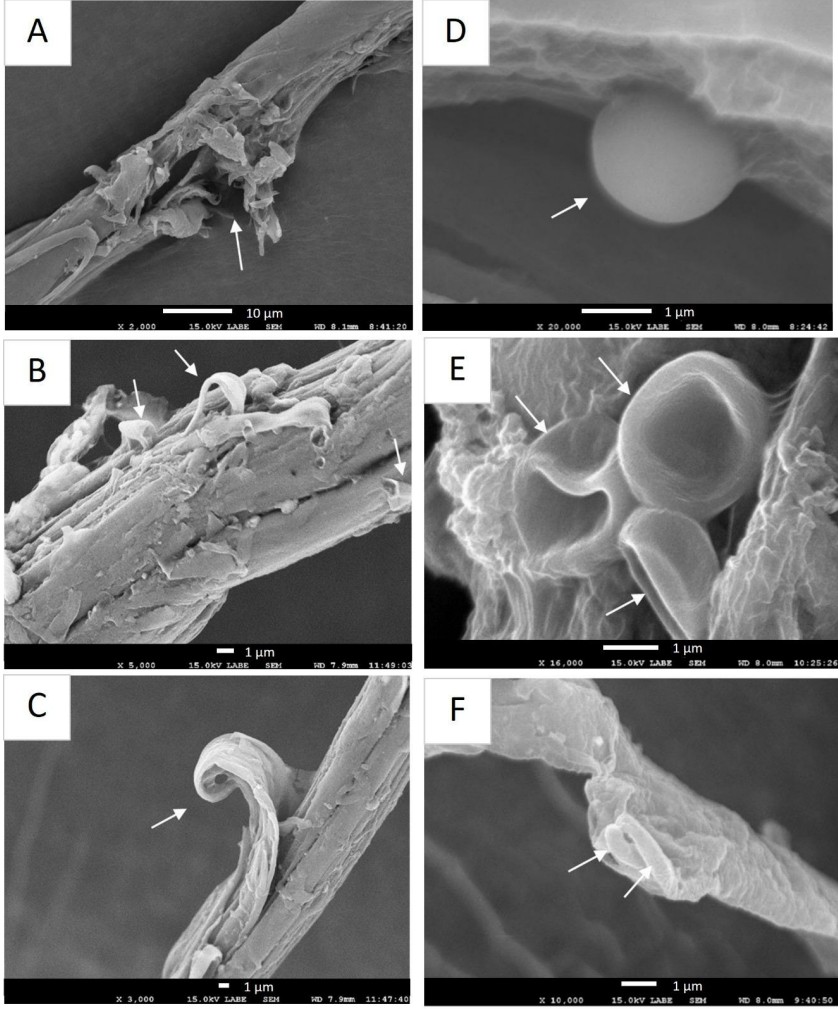

**Figure 8.** Scanning electron microscope images of microfibers attached to *Sargassum* exhibiting various features. The arrows in the figures point at: (**A**) wear, transverse ruptures, and cavities; (**B,C**) wastage and sheet detachment; (**D,E**) object resembling Coccoid bacteria; and (**F**) object resembling Bacillar bacteria. Images (**C,D,F**) were captured at 15 kV LABE.

## 4. Discussion

The findings of this study highlight the extensive presence of microfibers in pelagic *Sargassum* reaching the Mexican Caribbean throughout all observed months. Transparent, blue, and black fibers were the predominant colors identified, aligning with the documented prevalence of microplastic forms and colors in the open ocean [52,53] and marine fauna [54,55]. The methodology employed in this study involved subjecting *Sargassum* to a 48 h treatment with 30% hydrogen peroxide. This process was repeated three times to partially remove organic matter and induce discoloration, making it easier to visualize MPs using a stereoscope. The observed fibers were identified as synthetic polymers because non-synthetic polymers, like cellulose, cotton, and silk, usually degrade during organic matter removal. The primary origin of synthetic fibers in oceans can be traced to mechanical and chemical stresses imposed on synthetic fabrics, ultimately entering the sea through wastewater discharge [56]. Additional sources include the abrasion of vehicle tires, the

degradation of cigarette filters, and the fragmentation of maritime equipment such as ropes and fishing nets [57,58].

The results suggest that *Sargassum fluitans* III exhibits a substantial retention capacity for microfibers. Other marine macrophytes have reported similar capacities [50,59]. For example, *Ulva prolifera*'s distinctive tubular body and highly branched thallus enable it to retain high concentrations of microplastics (MPs) [60,61]. Recent studies also indicate that seagrasses can elevate MP abundance in seagrass bed sediments by 2.1 to 2.9 times [62]. Further research is essential to elucidate the characteristics that facilitate the accumulation of MPs in pelagic *Sargassum*.

In the present study, the quantity of MPs in *Sargassum* ranged from 1500 to 17,900 items/kg dry weight. These values surpass those reported by [19] for the Mexican Caribbean coast in 2022 (11.2–19.3 items/kg fresh weight). Considering *Sargassum*'s elevated moisture content (82–95%; [63]), the MP concentrations reported in the present study for dry samples would equate to approximately 150–1790 items/kg in fresh samples, significantly exceeding the figures reported by [19]. The notable disparities observed between the two studies, conducted on the same coast with only a one-year gap, may be attributed to methodological variations or highlight a high temporal variability in MP abundance associated with *Sargassum*. It is important to note that the abundance of MPs could even be higher, as some MPs could have been overlooked during optical identification, and others could have become detached from the algae during the drying process.

The MP concentration found in recently beach-cast *Sargassum* spp. in the Mexican Caribbean is more similar to that documented for *Ulva prolifera* in the Yellow Sea in August 2019 (7358.2 ± 1848.5 items/kg DW) and June 2020 (2002.9 ± 331.5 items/kg DW) by [64]. It is also comparable to that found in beach sediments in the Lesser Antilles (261–620 items/kg; [65]), in Guanabara Bay, Brazil (160–1000 items/kg; [40]), in Colombia's central Caribbean coast (557–2457 items/kg; [66]), and in the Amazon River basin, Ecuador (0–4200 items/kg; [67]).

Given that an estimated 10,105 to 40,932 cubic meters of *Sargassum* were removed per kilometer of the beach during 2018–2019 and 2021–2022 in the northern Mexican Caribbean [68], the potential influx of MPs to this coast could be considerable. Using the volume-mass conversion factor provided by [69], equating one cubic meter to 276 kg leads to annual *Sargassum* landings ranging from 2789 to 11,297 tons per kilometer. Consequently, this suggests that there could have been between $0.1 \times 10^9$ and $17.3 \times 10^9$ microplastics adhered to the *Sargassum* that annually reached each kilometer of the beach or between 41,400 and 494,040 items per cubic meter.

Currently, insufficient information is available to determine if the MPs attached to *Sargassum* reaching the Mexican Caribbean coast became entrapped in the Atlantic Ocean, the Caribbean Sea, or the coastal waters in front of the Mexican Caribbean. It is plausible that this entrapment occurred in all three regions during the months *Sargassum* mats traveled from West Africa to the Mexican Caribbean coast.

Countries in Latin America and the Caribbean exhibit poor regulations for MP legislation and mitigation. In more developed countries, MP mitigation strategies primarily focus on preventing their release into the environment through wastewater treatment plants [70]. However, in Latin America and the Caribbean, nearly 40 million people lack access to essential waste collection services, and approximately 17,000 tons of plastic waste are disposed of in open dumpsites daily [71]. Moreover, in many countries, wastewater is still directly discharged into water bodies without proper treatment [72].

Only approximately 10% of the *Sargassum* reaching Mexico is actively removed from the beaches by entities such as hotels, municipalities, and marines [45]. The rest undergoes natural decomposition, returning to the sea during storms or high-tide episodes. Therefore, the effective removal of beach-cast *Sargassum* to designated landfills holds the potential for simultaneously extracting MPs attached to it from ocean and coastal waters. Failure to implement such measures would allow *Sargassum* to function as a vector for MPs in coastal ecosystems, posing risks to diverse organisms across various trophic levels in coral reefs,

seagrass beds, beaches, and mangroves. These organisms may ingest MPs through direct consumption or indirectly through prey or respiration, resulting in physical and chemical impacts [73]. MPs could accumulate in animal organs, causing mechanical obstructions [74]. Furthermore, the associated chemical additives may bioaccumulate within each trophic level [49] and potentially transfer to higher trophic levels [27]. Consequently, exposure to elevated concentrations of MPs could disrupt coastal habitat assemblages [75].

Should *Sargassum* blooms persist, the influx of MPs into coastal ecosystems in affected countries may intensify over time. The situation could be further exacerbated by blooms of other drifting macroalgae, which possess diverse mechanisms for trapping [60] and concentrating plastic debris from the surrounding water [64]. At present, plastic pollution has been documented in various species of macroalgae, including *Ulva prolifera*, *U. lactuca*, *Sargassum horneri*, *Gracilaria lemaneiformis*, *Chondrus ocellatus*, and *Saccharina japonica* [61,76]. This underscores the broader impact of macroalgae blooms on the transportation and accumulation of MPs along coastlines. The possible presence of bacteria on MPs adhered to *Sargassum* also underscores their potential role as carriers or vectors of pathogens across the ocean [77] and represents a health risk for humans and other species in various ecosystems affected by *Sargassum*'s massive influxes.

Moreover, the presence of MPs in *Sargassum* can compromise its suitability as a raw ingredient in animal feed, fertilizer, or compost for agricultural purposes. However, this does not necessarily mean that pelagic *Sargassum* species cannot be industrialized. Instead, its derivatives need to be carefully analyzed for potential applications. *Sargassum* species have demonstrated promise for various uses. For example, phytochemicals from *S. wightii* were found to be effective against the mosquito *Aedes aegypti* [78], extracts from *S. linearifolium* exhibited antioxidant, antimicrobial, and anticancer activities [79], and those from *S. horneri* bio-stimulated the growth of the red algae *Neopyropia yezoensis* [80]. Additionally, hydrochar obtained from *S. muticum* demonstrated a high capacity to remove pollutants from water [81]. The valorization of pelagic *Sargassum* is essential to offset the elevated cleanup cost and to mitigate ecological, economic, and human health-related impacts.

## 5. Conclusions

This study demonstrates that microplastics, especially fibers, pollute pelagic *Sargassum* landing in the Mexican Caribbean. The large quantity of MPs found signifies the potential role of these macroalgae as conduits for the transfer of MPs from oceanic to coastal ecosystems. The import of MPs can significantly impact coastal ecosystems like coral reefs, seagrass beds, beaches, lagoons, and mangroves. An efficient cleanup of beach-cast *Sargassum* spp. from the Mexican Caribbean could remove between $0.1 \times 10^9$ and $17.3 \times 10^9$ MPs/km/year, offering a promising mitigation strategy for plastic pollution in the examined marine environment.

**Author Contributions:** Conceptualization, T.P.G.C. and D.A.A.; methodology, T.P.G.C., D.A.A., V.C.E. and R.E.R.-M.; software, T.P.G.C.; validation, T.P.G.C., D.A.A., V.C.E. and R.E.R.-M.; formal analysis, T.P.G.C., D.A.A. and V.C.E.; investigation, T.P.G.C., D.A.A. and V.C.E.; resources, D.A.A.; data curation, T.P.G.C. and V.C.E.; writing—original draft preparation, D.A.A., T.P.G.C. and R.E.R.-M.; writing—review and editing, D.A.A., T.P.G.C., V.C.E. and R.E.R.-M.; visualization, T.P.G.C. and R.E.R.-M.; supervision, D.A.A.; project administration, D.A.A.; funding acquisition, D.A.A. All authors have read and agreed to the published version of the manuscript.

**Funding:** This research was funded by CINVESTAV-IPN, Laboratorio de Biología Marina. A research grant scholarship (Number CVU1079631) was provided by Consejo Nacional de Ciencia y Tecnología (CONACYT) to T.P.G.C.

**Informed Consent Statement:** Not applicable.

**Data Availability Statement:** The data used in this study are available at https://github.com/rerodriguezmtz/MPinSargassum (accessed on 20 February 2024).

**Acknowledgments:** We are grateful to Victor Rejón Moo for his support in analyzing microplastics with SEM and EDS in the Laboratorio Nacional de Nano y Biomateriales, CINVESTAV-IPN, and MSc. Silvia Granados Puerto for providing the materials to conduct this study. We also thank the Servicio Académico de Monitoreo Meteorológico y Oceanográfico (SAMMO) from the Universidad Nacional Autónoma de México for providing the data on air temperature and wind direction.

**Conflicts of Interest:** The authors declare no conflicts of interest. The funders had no role in the study's design; in the collection, analyses, or interpretation of data; in the writing of the manuscript; or in the decision to publish the results.

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
