# Peer review of "Pelagic Sargassum as a Potential Vector for Microplastics into Coastal Ecosystems"

_phycology, doi:10.3390/phycology4010008_

Round 1

Reviewer 1 Report

Comments and Suggestions for Authors

This paper is clearly presented and is a useful contribution to the literature on the Sargassum/golden tides issue. It's particular value is highlighting the potential for MP pollution and the importance of this, in addition to the heavy metal accumulation, to be fully taken into account when either disposal of Sargassum or economically viable end uses are considered.  

Author Response

Pelagic Sargassum as a potential vector for microplastics into coastal ecosystems

Manuscript ID: phycology-2800236

We thank the reviewers for the comments made to the manuscript which helped us to improve it. Just to let you know, all recommendations were attended. Our answers to each comment are shown in blue.

Reviewer 1

This paper is clearly presented and is a useful contribution to the literature on the Sargassum/golden tides issue. It's particular value is highlighting the potential for MP pollution and the importance of this, in addition to the heavy metal accumulation, to be fully taken into account when either disposal of Sargassum or economically viable end uses are considered.

We thank Reviewer 1 for this comment.

Reviewer 2 Report

Comments and Suggestions for Authors

Manuscript “Pelagic Sargassum as a vector for microplastics into coastal ecosystems” of Dalila Aldana Arana, Tania P. Gil Cortés, Víctor Castillo Escalante, and Rosa E. Rodríguez-Martínez is the result of a study of microplastics on macroalgae sampled at one point over 1 year. The problem of microplastic pollution in various water bodies attracts the attention of scientists around the world and research on this topic is currently very popular. Due to the novelty of the topic and its worldwide significance, the relevance of the presented research does not need confirmation, however, in connection with the above, it requires a very careful and precise approach. In my opinion, the authors failed to cope with this.

The main complaint is that the results are described and presented extremely unsatisfactorily, and the discussion does not explain the results at all, although there is something to talk about. Based on the description of materials and methods, the results expected to see not only data on the number of particles at different times of the year and their color, but also how the color data correlated with the morphology of the particles. It is unclear whether there is any difference in the presence of particles on stems or leaves, or whether there is a difference in the distribution of washed particles and on the algae themselves. The number of photographs presented is unsatisfactory - there are not all images of microplastic particles of primary colors. There is no date of sampling in Table S1, and the elemental analysis appears to have been done separately from the rest of the study. The discussion is a literature review and a popular science article on the problem of microplastic transport by Sargassum from the open sea to the coast, and all this has little to do with the results obtained.

General comments:

In the introduction, in connection with the results obtained, it is worth clarifying the description of microplastics - what is the morphology associated with, what is affected by the color of the particles, and a more detailed and specific description of the potential harm of the particles themselves, and not just what they can carry. In the last sentence of the introduction, it is better to clarify with what methods the research was conducted.

In Materials and Methods, the first paragraph (lines 74-80) is very similar to the discussion section. When describing the study area, it was enough to indicate that on this part of the coast there is a significant mass of sargassum washed ashore. Much of the description should concern the temperature and other physical characteristics during the sampling period, as well as a description of the specific hydrological conditions, streams and winds, especially if they vary seasonally. This will then allow us to speculate about which areas the algae were brought from and what could have influenced the concentration of microplastics in the samples. The map (Fig. 1) should at least symbolically represent the streams around the studied area. When describing sampling, a table is required that would reflect the date of sampling, the exact weight of the sample taken, wet and dry, and other characteristics.

Lines 102-106 is not sample collection, but sample treatment.

Lines 122-125 are also discussion and not exact description of materials, because the authors justify the use of the SEM method.

The description of the results has already been mentioned above. At the same time, there is no clear data (except for a small mention on lines 158-159) about which particles predominated in morphology at different times of the year.

In the discussion, an entire paragraph is devoted to a story about the composition of microplastics (lines 213-220), but the results contain only elemental analysis, which cannot be used to detect whether it is PVC or nylon. It is possible that the quality of the microplastic is reflected by its color and morphology, however, there is no mention of this in any chapter of the manuscript.

Overall, despite the relevance of the topic and the interesting idea that sargassum can accumulate and transport microplastics, the study gives the impression of a draft. The topic of microplastic pollution is highly speculative thanks to studies like this one. I would not like this topic to become a populist slogan due to carelessly presented results. To avoid this, the results must be described carefully and in detail, and all techniques must be explained.

Author Response

Pelagic Sargassum as a potential vector for microplastics into coastal ecosystems

Manuscript ID: phycology-2800236

We thank the reviewers for the comments made to the manuscript which helped us to improve it. All recommendations were attended. Our answers to each comment are shown in blue.

Reviewer 2

Manuscript “Pelagic Sargassum as a vector for microplastics into coastal ecosystems” of Dalila Aldana Arana, Tania P. Gil Cortés, Víctor Castillo Escalante, and Rosa E. Rodríguez-Martínez is the result of a study of microplastics on macroalgae sampled at one point over 1 year. The problem of microplastic pollution in various water bodies attracts the attention of scientists around the world and research on this topic is currently very popular. Due to the novelty of the topic and its worldwide significance, the relevance of the presented research does not need confirmation, however, in connection with the above, it requires a very careful and precise approach. In my opinion, the authors failed to cope with this.

The main complaint is that the results are described and presented extremely unsatisfactorily, and the discussion does not explain the results at all, although there is something to talk about.

The results and discussion sections have been modified in response to the comments made by the reviewers.

Based on the description of materials and methods, the results expected to see not only data on the number of particles at different times of the year and their color, but also how the color data correlated with the morphology of the particles.

We have revised the figure illustrating the color relative abundance per month corresponding to each MP type. It is important to note that the morphology of the particles may not necessarily correlate with their color. According to existing literature, the coloration is linked to the plastic's origin. In light of this, we have incorporated pertinent information and references into the discussion to provide a more comprehensive understanding.

It is unclear whether there is any difference in the presence of particles on stems or leaves, or whether there is a difference in the distribution of washed particles and on the algae themselves.

In this study, the presence of microplastics (MP) on stems or leaves is deemed irrelevant, as our primary focus is on determining the overall abundance rather than elucidating the entrapment mechanisms. Enumeration of MP was conducted in the blades, stems, and the rinsing liquid. Including information in the text regarding the substantial presence of MP on Sargassum post-washing serves two main purposes. Firstly, it prompts future studies to recognize that measuring MP solely in the rinsing liquid might lead to underestimating their abundance. Secondly, it is relevant for potential applications and disposal considerations of these algae, as emphasized by Reviewer 1. This information has been integrated into the discussion section.

The number of photographs presented is unsatisfactory - there are not all images of microplastic particles of primary colors.

The Figure was modified to include more MP colors.

There is no date of sampling in Table S1, and the elemental analysis appears to have been done separately from the rest of the study.

The sampling date was added to Table S1. The elemental analysis was done on Sargassum samples collected within the study's time frame (March to October 2021 and January 2022).

The discussion is a literature review and a popular science article on the problem of microplastic transport by Sargassum from the open sea to the coast, and all this has little to do with the results obtained.

We thank the reviewer for the comments. The Results and Discussion sections were restructured. Additional information and references have been included.

General comments:

In the introduction, in connection with the results obtained, it is worth clarifying the description of microplastics - what is the morphology associated with, what is affected by the color of the particles, and a more detailed and specific description of the potential harm of the particles themselves, and not just what they can carry.

The introduction section was modified according to the comments of all reviewers.

In the last sentence of the introduction, it is better to clarify with what methods the research was conducted.

We respectfully differ from the reviewer's perspective on including the methods in the final sentence of the introduction, given that the pertinent information is already presented in the Methodology section. Nevertheless, we have incorporated details about the protocol employed for pretreating the macroalgae for microplastic extraction in the abstract.

In Materials and Methods, the first paragraph (lines 74-80) is very similar to the discussion section. When describing the study area, it was enough to indicate that on this part of the coast there is a significant mass of sargassum washed ashore.

We believe that offering readers information about the volumes of Sargassum landings is crucial for comprehending the issue's magnitude. Merely stating that "masses are significant" lacks context. Given the considerable variability in Sargassum landing volumes across different locations, we find it more informative to provide a specific quantity.

Much of the description should concern the temperature and other physical characteristics during the sampling period, as well as a description of the specific hydrological conditions, streams and winds, especially if they vary seasonally. This will then allow us to speculate about which areas the algae were brought from and what could have influenced the concentration of microplastics in the samples. The map (Fig. 1) should at least symbolically represent the streams around the studied area. When describing sampling, a table is required that would reflect the date of sampling, the exact weight of the sample taken, wet and dry, and other characteristics.

Figure 1 was modified to include a diagram of the main currents involved in the transportation of pelagic Sargassum to the Mexican Caribbean coast (South Equatorial Current, the Caribbean Current, and the Yucatan Current).

Lines 102-106 is not sample collection, but sample treatment.

The subtitle was modified to “Sample treatment”

Lines 122-125 are also discussion and not exact description of materials, because the authors justify the use of the SEM method.

The sentence was removed from the paragraph.

The description of the results has already been mentioned above. At the same time, there is no clear data (except for a small mention on lines 158-159) about which particles predominated in morphology at different times of the year.

The text was modified. A new figure was incorporated to show the temporal variability of MP morphologies

In the discussion, an entire paragraph is devoted to a story about the composition of microplastics (lines 213-220), but the results contain only elemental analysis, which cannot be used to detect whether it is PVC or nylon. It is possible that the quality of the microplastic is reflected by its color and morphology, however, there is no mention of this in any chapter of the manuscript.

The section mentioned by the reviewer was removed from the new version of the manuscript

Overall, despite the relevance of the topic and the interesting idea that sargassum can accumulate and transport microplastics, the study gives the impression of a draft. The topic of microplastic pollution is highly speculative thanks to studies like this one. I would not like this topic to become a populist slogan due to carelessly presented results. To avoid this, the results must be described carefully and in detail, and all techniques must be explained.

In the revised version of the manuscript, we have provided a more detailed explanation of the methods and results. However, we, as the authors, find the severe disqualification made by the reviewer in this comment on our work to be unacceptable and disrespectful. We are open to making any necessary changes to enhance the manuscript, but we kindly request a respectful discourse among peers.

Reviewer 3 Report

Comments and Suggestions for Authors

Review - Pelagic Sargassum as a vector for microplastics into coastal ecosystems

This study quantifies and characterizes the microplastics associated with pelagic Sargassum at a single sampling location on the Mexican Caribbean coast. This is an interesting study. However, additional detail is required in the Methods and Results. Additionally, the Discussion needs further development to consider the origins and impacts of the observed microplastics.

Abstract -

Line 14 – Remove the comma before ‘affecting’.

Lines 18-19 – The ‘fiber colors. . . ‘ sentence is confusing. Rework.

Line 21 – The Abstract does not provide any results demonstrating consistent presence of plastic pollution in Sargassum. Perhaps add a sentence that indicates the % of samples that contained plastic.

Line 24 – But then the plastics are transported to the ‘designated sites’. Are these landfills or is the Sargassum being transported for other uses, thus bringing the plastic into that process?

Introduction –

Excellent background overview with appropriate citations.

Methods –

Lines 81-87 – Good summary of the precipitation patterns, I hope that you will address the potential for the plastics to have been introduced from the Mexican coast and coastal waters.

Line 94 – Suggestion to change “freshly gathered” to “fresh”. At present, the “freshly gathered” does capture the intent of the clause.

Line 95 – How were the samples “manually collected”? Were they scooped by hand or with a bucket? How was it ensured that microplastics did not fall off of the samples during collection or that microplastics from the water were not collected?

Line 99 – Upon what surface were the samples sun dried? How was it ensured that microplastics did not fall off of the samples during drying or that microplastics were not added from the environment during the drying time?

Line 102 – How were the 5g subsamples selected from the bulk sample? Why wasn’t the entire sample processed?

Line 103 – Please include an explanation of the purpose and a citation for the triplicate 48 hr hydrogen peroxide treatments.

Line 128 – Be clear about how many particles were examined by SEM. Are the SEM images of particles from particles on Sargassum or free particles from the liquid?

Results –

Line 145 – Please provide some initial descriptive statistics. What was the range of quantities of microplastic particles observed in each 5g subsample? While the minimum and maximum size limits for microplastic observation were mentioned in the methods, it would be helpful to include a characterization of the observed sizes of the particles encountered. How long were fibers? What was the longest diameter of fragments?

Figure 1 – In seeing this figure, I realize that there needs to be a better explanation of the Methods. Was the hydrogen peroxide digestion run three times on a single subsample from each originally collected sample (that is how the Methods currently read) or was the digestion run a single time on three different subsamples from the originally collected sample? There was no replication upon collection, correct?

Figure 4 – Suggestion to replace “leaf” with “blade” throughout the caption to correspond with the macroalgal terminology of stipe.

Line 190 – Confusing statement: “calcifying epibionts like diatoms.” Confusing to attribute the presence of calcium to diatoms and not mention the connection between diatoms and silicon.

Discussion –

Line 225 – Given the trajectory of the Discussion it seems important to include a comparison to microplastic concentrations estimated for coastal waters (without Sargassum). Do these data exist in the literature? If not, the topic is still worth mentioning.

Line 231 – What was the lower size limit for microplastics examined in [19]? It has been observed that because of fragmentation processes, abundance increases with decrease in microplastic size.

Line 261 – Given waste management conditions in the Caribbean, it is critical to include a discussion of the potential for the plastics to have originated locally. This needs to be offered as a counter point to the argument that Sargassum inundations are introducing tons of plastic to the local system.

Conclusions –

Lines 279-280 – Statement about the “role of these macroalgae in the Atlantic Ocean as conduits for the transfer of MP from oceanic to coastal ecosystems” seems a inappropriate, given that the study does not measure or discuss the origins of the microplastic particles found adhered to pelagic Sargassum.

Lines 281-282 – Please elaborate earlier on the potential impacts of microplastics of this size range on the ecosystems highlighted. What are the specific, measured impacts of microplastic fibers of this size range on corals and seagrass, mangrove, beach and lagoon communities of organisms?

Author Response

Pelagic Sargassum as a potential vector for microplastics into coastal ecosystems

Manuscript ID: phycology-2800236

We thank the reviewers for the comments made to the manuscript which helped us to improve it. All recommendations were attended. Our answers to each comment are shown in blue.

Reviewer 3

This study quantifies and characterizes the microplastics associated with pelagic Sargassum at a single sampling location on the Mexican Caribbean coast. This is an interesting study. However, additional detail is required in the Methods and Results. Additionally, the Discussion needs further development to consider the origins and impacts of the observed microplastics.

Additional details were incorporated in the Methods and Results section. The discussion was restructured.

Abstract -

Line 14 – Remove the comma before ‘affecting’.

Done

Lines 18-19 – The ‘fiber colors. . . ‘ sentence is confusing. Rework.

The sentences was improved

Line 21 – The Abstract does not provide any results demonstrating consistent presence of plastic pollution in Sargassum. Perhaps add a sentence that indicates the % of samples that contained plastic.

The following sentence was added “ All samples analyzed contained MP”

Line 24 – But then the plastics are transported to the ‘designated sites’. Are these landfills or is the Sargassum being transported for other uses, thus bringing the plastic into that process?

The sentence was modified, changing “designated sites” to “landfills”

Introduction –

Excellent background overview with appropriate citations.

Methods –

Lines 81-87 – Good summary of the precipitation patterns, I hope that you will address the potential for the plastics to have been introduced from the Mexican coast and coastal waters.

Additional information and references were included in the discussion to present the high percentage of untreated wastewater that ends up on the Mexican Caribbean coast and may be one of the sources of MP pollution.

Line 94 – Suggestion to change “freshly gathered” to “fresh”. At present, the “freshly gathered” does capture the intent of the clause.

The text was changed according to the recommendation of the Reviewer.

Line 95 – How were the samples “manually collected”? Were they scooped by hand or with a bucket? How was it ensured that microplastics did not fall off of the samples during collection or that microplastics from the water were not collected?

The text was modified as follows:

Fresh Sargassum fluitans III, was meticulously collected by hand from specimens floating near the shore or those recently washed ashore in proximity to the Unidad Académica de Sistemas Arrecifales, Instituto de Ciencias del Mar y Limnología, UNAM (Figure 1b). The collected samples were promptly placed into a previously washed plastic box and allowed to sundry. Upon complete drying, the entire box contents were carefully transferred to a Ziploc bag to prevent the loss of any particulate matter that might have dislodged during the process.

Line 99 – Upon what surface were the samples sun dried? How was it ensured that microplastics did not fall off of the samples during drying or that microplastics were not added from the environment during the drying time?

As mentioned above, the samples were immediately placed in a previously washed plastic box and left to sundry. During the dry time an aluminum paper cover the box to avoid environmental contamination. This information has been included in the manuscript.

Line 102 – How were the 5g subsamples selected from the bulk sample? Why wasn’t the entire sample processed?

The 5g subsamples were selected randomly. Entire samples were not processed because the abundance of MP in Sargassum is considerable, and there were time and personnel constraints. Subsampling enabled us to collect meaningful data ensuring a manageable workload.

Line 103 – Please include an explanation of the purpose and a citation for the triplicate 48 hr hydrogen peroxide treatments.

This line was removed, and the method was explained more clearly.

Line 128 – Be clear about how many particles were examined by SEM. Are the SEM images of particles from particles on Sargassum or free particles from the liquid?

The referred text was removed from the manuscript as the sample size was too small.

Results –

Line 145 – Please provide some initial descriptive statistics. What was the range of quantities of microplastic particles observed in each 5g subsample? While the minimum and maximum size limits for microplastic observation were mentioned in the methods, it would be helpful to include a characterization of the observed sizes of the particles encountered. How long were fibers? What was the longest diameter of fragments?

The descriptive statistics have been included in the results section. MP were not measured.

Figure 1 – In seeing this figure, I realize that there needs to be a better explanation of the Methods. Was the hydrogen peroxide digestion run three times on a single subsample from each originally collected sample (that is how the Methods currently read) or was the digestion run a single time on three different subsamples from the originally collected sample? There was no replication upon collection, correct?

It has now been specified that three samples were analyzed from each monthly sample and that the digestion was run once on each.

Figure 4 – Suggestion to replace “leaf” with “blade” throughout the caption to correspond with the macroalgal terminology of stipe.

Done

Line 190 – Confusing statement: “calcifying epibionts like diatoms.” Confusing to attribute the presence of calcium to diatoms and not mention the connection between diatoms and silicon.

The statement: “calcifying epibionts like diatoms” was removed

Discussion –

Line 225 – Given the trajectory of the Discussion it seems important to include a comparison to microplastic concentrations estimated for coastal waters (without Sargassum). Do these data exist in the literature? If not, the topic is still worth mentioning.

Additional information on MP contamination in the Caribbean has been incorporated. There is no data on the specific analysis of untreated wastewater in the Mexican Caribbean, but there is information that more than 80% of the population discharges its wastewater without any treatment. Besides the attachment of MP is influenced by various factors, including available surface area for contact, algal structure, and wall polysaccharides, among others.

Line 231 – What was the lower size limit for microplastics examined in [19]? It has been observed that because of fragmentation processes, abundance increases with decrease in microplastic size.

The size of the MP analyzed by [19] was less than or equal to 5 mm in any of their dimensions.

Line 261 – Given waste management conditions in the Caribbean, it is critical to include a discussion of the potential for the plastics to have originated locally. This needs to be offered as a counter point to the argument that Sargassum inundations are introducing tons of plastic to the local system.

We now mention in the manuscript that “Currently, insufficient information is available to determine if the MP attached to Sargassum reaching the Mexican Caribbean coast became entrapped in the Atlantic Ocean, the Caribbean Sea, or the coastal waters in front of the Mexican Caribbean. It is plausible that this entrapment occurred in all three regions during the months Sargassum mats traveled from West Africa to the Mexican Caribbean coast.”

Conclusions –

Lines 279-280 – Statement about the “role of these macroalgae in the Atlantic Ocean as conduits for the transfer of MP from oceanic to coastal ecosystems” seems a inappropriate, given that the study does not measure or discuss the origins of the microplastic particles found adhered to pelagic Sargassum.

The sentence was modified to mention that Sargassum has a potential role for the transport of MP. Additional information was added about the low percentage of wastewater treatment on the Caribbean coast of Mexico, which is one of the possible causes of MP in Sargassum.

Lines 281-282 – Please elaborate earlier on the potential impacts of microplastics of this size range on the ecosystems highlighted. What are the specific, measured impacts of microplastic fibers of this size range on corals and seagrass, mangrove, beach and lagoon communities of organisms?

We added information and references regarding impacts of MP various Caribbean coastal ecosystems.

Reviewer 4 Report

Comments and Suggestions for Authors

Dear Authors,

I have carefully reviewed your manuscript titled "Pelagic Sargassum as a vector for microplastics into coastal ecosystems".

Overall, I find the paper well-written and commend your efforts in shedding light on an important aspect of macroalgal blooms. The study significantly contributes to our understanding of the influence of pelagic Sargassum on microplastic distribution along coastal regions.

I would like to commend the idea of extending this research to other macroalgae species and comparing their interactions with microplastics. The attachment of microplastics is influenced by various factors, including available surface area for contact, algal structure, and wall polysaccharides, among others. Exploring these aspects further could contribute significantly to the field.

In conclusion, I recommend a "minor revision" for your manuscript (all suggestions are in the attached pdf). The suggested changes aim to enhance the clarity and depth of your findings. I appreciate the valuable contribution your work makes to the understanding of macroalgal blooms and their association with microplastics.

Regards

Author Response

Pelagic Sargassum as a potential vector for microplastics into coastal ecosystems

Manuscript ID: phycology-2800236

We thank the reviewers for the comments made to the manuscript which helped us to improve it. All recommendations were attended. Our answers to each comment are shown in blue.

Reviewer 4

Dear Authors,

I have carefully reviewed your manuscript titled "Pelagic Sargassum as a vector for microplastics into coastal ecosystems".

Overall, I find the paper well-written and commend your efforts in shedding light on an important aspect of macroalgal blooms. The study significantly contributes to our understanding of the influence of pelagic Sargassum on microplastic distribution along coastal regions.

I would like to commend the idea of extending this research to other macroalgae species and comparing their interactions with microplastics. The attachment of microplastics is influenced by various factors, including available surface area for contact, algal structure, and wall polysaccharides, among others. Exploring these aspects further could contribute significantly to the field.

Thank you.

In conclusion, I recommend a "minor revision" for your manuscript (all suggestions are in the attached pdf). The suggested changes aim to enhance the clarity and depth of your findings. I appreciate the valuable contribution your work makes to the understanding of macroalgal blooms and their association with microplastics.

Comments in PDF file

Ln 43-46 - Sargassum as a raw material and their derivatives (extracts) are being tested from many points of view, add some other uses, they are not just a risky resource, here some citations to add:

Thank you very much for the information and the very complete list of other uses/products/properties from Sargassum spp.

Abu-Khudir, R., Ismail, G. A., & Diab, T. (2021). Antimicrobial, Antioxidant, and Anti-Tumor Activities of Sargassum linearifolium and Cystoseira crinita from Egyptian Mediterranean Coast. Nutrition and Cancer, 73(5), 829–844. Scopus. https://doi.org/10.1080/01635581.2020.1764069

Achary, A., Muthalagu, K., & Guru, M. S. (2014). Identification of phytochemicals from SArgassum wightii against Aedes aegypti. International Journal of Pharmaceutical Sciences Review and Research, 29(1), 314–319. Scopus.

Han, S., Park, J.-S., Umanzor, S., Yarish, C., & Kim, J. K. (2022). Effects of extraction methods for a new source of biostimulant from Sargassum horneri on the growth of economically important red algae, Neopyropia yezoensis. Scientific Reports, 12(1), 11878. https://doi.org/10.1038/s41598-022-16197-0

Rasam, S., Talebkeikhah, F., Talebkeikhah, M., Salimi, A., & Moraveji, M. K. (2021). Physico-chemical properties prediction of hydrochar in macroalgae Sargassum horneri hydrothermal carbonisation. International Journal of Environmental Analytical Chemistry, 101(14), 2297–2318. https://doi.org/10.1080/03067319.2019.1700973

Spagnuolo, D., Bressi, V., Manghisi, A., Azzara’, M., Teresa Chiofalo, M., Espro, C., Genovese, G., Morabito, M., & Trifilo’, P. (2022). Influence of aqueous phase from hydrothermal carbonization (AHL) of Sargassum muticum (Phaeophyceae) on germination and growth of Phaseolus vulgaris (Fabaceae).

Spagnuolo, D., Iannazzo, D., Len, T., Balu, A. M., Morabito, M., Genovese, G., Espro, C., & Bressi, V. (2023). Hydrochar from Sargassum muticum: A sustainable approach for high-capacity removal of Rhodamine B dye. RSC Sustainability, 10.1039.D3SU00134B. https://doi.org/10.1039/D3SU00134B

The text was modified and many of the references were included

Lns 50-52 - add something more regarding the trapping of marine organisms and the accumulation of plastics and microplastics:

Thank you for the references. We incorporated several of them in the new version of the manuscript.

Dahl, M., Bergman, S., Björk, M., Diaz-Almela, E., Granberg, M., Gullström, M., Leiva-Dueñas, C., Magnusson, K., Marco-Méndez, C., Piñeiro-Juncal, N., & Mateo, M. Á. (2021). A temporal record of microplastic pollution in Mediterranean seagrass soils. Environmental Pollution, 273. Scopus. https://doi.org/10.1016/j.envpol.2021.116451

Li, Q., Su, L., Ma, C., Feng, Z., & Shi, H. (2022). Plastic debris in coastal macroalgae. Environmental Research, 205, 112464. https://doi.org/10.1016/j.envres.2021.112464

Mancuso, M., Genovese, G., Porcino, N., Natale, S., Crisafulli, A., Spagnuolo, D., Catalfamo, M., Morabito, M., & Bottari, T. (2023). Psammophytes as traps for beach litter in the Strait of Messina (Mediterranean Sea). Regional Studies in Marine Science, 65, 103057. https://doi.org/10.1016/j.rsma.2023.103057

Ng, K. L., Suk, K. F., Cheung, K. W., Shek, R. H. T., Chan, S. M. N., Tam, N. F. Y., Cheung, S. G., Fang, J. K.-H., & Lo, H. S. (2022). Macroalgal morphology mediates microplastic accumulation on thallus and in sediments. Science of the Total Environment, 825. Scopus. https://doi.org/10.1016/j.scitotenv.2022.153987

Zhang, T., Wang, J., Liu, D., Sun, Z., Tang, R., Ma, X., & Feng, Z. (2022). Loading of microplastics by two related macroalgae in a sea area where gold and green tides occur simultaneously. Science of the Total Environment, 814. Scopus. https://doi.org/10.1016/j.scitotenv.2021.152809

Ln 88 – Improve quality of image 1b

The quality of the image was improved

Ln 94 - macroalgae are very plastic, it seems really difficult to think that it can be distinguished thanks to the "golden hue", one can assume this is the case, but without a DNA analysis the exact lineage cannot be confirmed. highlight this aspect. as you described below, write in advance here

Species ID was not done by the color. We use the “golden hue color” to distinguish between fresh and old Sargassum, which turns dark brown. We modified the text to avoid confusion.

Only two pelagic Sargassum species land on the Mexican Caribbean coast. S. natans (forms I and VIII) and S. fluitans (form III), and it is easy to differentiate among them based on the size of the blades and the presence of a spine in the vesicle.

Ln 304 - The link is not working (github)

The https address was modified.

Round 2

Reviewer 2 Report

Comments and Suggestions for Authors

I thank the authors of the manuscript for their attention to the comments made and the work done. In its present form, with a clear description of the methods and location of sample collection, the study has become clearer, and the changes made in the introduction and discussion allow to appreciate the work and conclusions of the authors. I regret being harsh in my remarks about the study. I take the field I have chosen very seriously, and it is difficult for me to restrain myself if I see signs of any falsification of science. Fortunately, in this case it seemed to me, which I am very happy about and wish the authors further success and new ideas in their work.

At the moment I am satisfied with the presentation of the results, as well as the quality and quantity of micrographs (and their size). However, there are a few minor fixes:

1. Regarding the title of paragraph 2.2., I meant exclusively the lines that I indicated. I think in general Paragraph 2.2. should be called Sample collection and treatment

2. The upper numbers in Figure 5 look cropped, please check the quality of the drawing.

Author Response

Pelagic Sargassum as a potential vector for microplastics into coastal ecosystems

Manuscript ID: phycology-2800236

Reviewer 2

I thank the authors of the manuscript for their attention to the comments made and the work done. In its present form, with a clear description of the methods and location of sample collection, the study has become clearer, and the changes made in the introduction and discussion allow to appreciate the work and conclusions of the authors. I regret being harsh in my remarks about the study. I take the field I have chosen very seriously, and it is difficult for me to restrain myself if I see signs of any falsification of science. Fortunately, in this case it seemed to me, which I am very happy about and wish the authors further success and new ideas in their work.

At the moment I am satisfied with the presentation of the results, as well as the quality and quantity of micrographs (and their size). However, there are a few minor fixes:

  1. Regarding the title of paragraph 2.2., I meant exclusively the lines that I indicated. I think in general Paragraph 2.2. should be called Sample collection and treatment

The title of the paragraph was changed to: “Sample collection and treatment”

  1. The upper numbers in Figure 5 look cropped, please check the quality of the drawing.

The figure was replaced

Reviewer 3 Report

Comments and Suggestions for Authors

Pelagic Sargassum as a potential vector for microplastics into coastal ecosystems

Manuscript ID: phycology-2800236

This study quantifies and characterizes the microplastics associated with pelagic Sargassum at a single sampling location on the Mexican Caribbean coast. The revisions sufficiently addressed many of the comments, but additional information is still needed in the methods and results.

Methods –

Line 129 – Change “sundry” to “sun dry”

Line 157 – How were the 5 g selected? In the prior paragraph it is stated that the team was careful to transfer everything from the sun drying bins to the ziplock bags, in case anything (e.g., plastics) fell off the Sargassum while drying. How were bits that may have fallen off accounted for in the subsample selection? Were the 5 g pulled directly from the whole dried Sargassum frond or a mixture of bits with this?

Line 142 – Change “underwent coverage” to “was covered by”

Line 151 – Because only optical identification and quantification was employed, it will be important in the Discussion to address that some plastics might have been missed

Results –

Lines 173-174 – Why does it matter the numbers of microplastics on Sargassum v in rinse water? It would be more useful to open the Results with the total found in 5g, whether in rinse or still attached to Sargassum.

Line 177 – Confusing sentence, with “of the monthly samples” on the end. Did 24-72% of the monthly samples still have microplastics on them or was it of the entire samples set? Or was it that 24-72% of the plastics remained on the samples?

Line 186 – Change “while” to “white”? Is it white or clear?

Discussion –

Lines 239-241 – This statement about some fibers potentially being non-synthetic calls into question the identification/verification of microplastics in this study. In fact, how do you know that the black fragment in Figure 3 is not a piece of tar adhered to the Sargassum?

Lines 256-257 – What were the methodological differences?

A discussion of the limitations to the selected visual methodology is missing but critical to include. Fragments and dark colors are most definitely easier to spot visually than fragments or films and transparent particles.

Additionally, the authors rebuttal notes that MP sizes were not recorded, but a size range would be a helpful characterization to understand the potential for ecological impact. There are size scales on the photos of particles. Were all particles photographed? Could the photos be analyzed to capture size information.

Author Response

Reviewer 3

This study quantifies and characterizes the microplastics associated with pelagic Sargassum at a single sampling location on the Mexican Caribbean coast. The revisions sufficiently addressed many of the comments, but additional information is still needed in the methods and results.

Methods –

Line 129 – Change “sundry” to “sun dry”

Done

Line 157 – How were the 5 g selected? In the prior paragraph it is stated that the team was careful to transfer everything from the sun drying bins to the ziplock bags, in case anything (e.g., plastics) fell off the Sargassum while drying. How were bits that may have fallen off accounted for in the subsample selection? Were the 5 g pulled directly from the whole dried Sargassum frond or a mixture of bits with this?

Sargassum thalli were placed on plastic trays for sun drying. The set of these fronds was transferred whole to the bags, and from these fronds, the three subsamples with a weight of 5 grams were taken in the laboratory. It is possible that some MP fell from sargassum in the ziplock bag, and thus, the quantity of MP reported could have been higher. This has now been mentioned in the discussion of the manuscript.

Line 142 – Change “underwent coverage” to “was covered by”

Done

Line 151 – Because only optical identification and quantification was employed, it will be important in the Discussion to address that some plastics might have been missed

In the discussion, we included the following paragraph: “In the present study, the quantity of MP in Sargassum ranged from 1,500 to 17,900 items/kg dry weight. It is important to note that the actual abundance might be higher, as there’s a possibility that some MP were overlooked during optical identification, and others could have become detached from the algae during the drying process.

Results –

Lines 173-174 – Why does it matter the numbers of microplastics on Sargassum v in rinse water? It would be more useful to open the Results with the total found in 5g, whether in rinse or still attached to Sargassum.

The paragraph was modified to “The monthly microplastics (MP) in Sargassum per sample (5 g) ranged from 16 to 98, resulting in monthly means fluctuating from 3.5 to 15.3 MP g-1 DW (Figure 3). The peak mean MP concentration was recorded in April 2021, while the lowest was observed in June 2021. It is worth noting that the relative abundance of MP that persisted on Sargassum after rinsing ranged from 24% to 72% of the monthly samples. This emphasizes that relying solely on rising the blades is insufficient for the complete removal of MP.”

Line 177 – Confusing sentence, with “of the monthly samples” on the end. Did 24-72% of the monthly samples still have microplastics on them or was it of the entire samples set? Or was it that 24-72% of the plastics remained on the samples?

As mentioned above it refers to MP remaining in Sargassum after the rinsing.

Line 186 – Change “while” to “white”? Is it white or clear?

We modified the sentence to avoid confusion to:

“Throughout the study period, fibers consistently constituted the predominant type of MP in Sargassum (91%), followed by fragments (8.3%); films (0.6%) and spheres (0.1%) were rare.”

Discussion –

Lines 239-241 – This statement about some fibers potentially being non-synthetic calls into question the identification/verification of microplastics in this study. In fact, how do you know that the black fragment in Figure 3 is not a piece of tar adhered to the Sargassum?

The paragraph was restructured: “The methodology employed in this study involved subjecting Sargassum to a 48-hour treatment with 30% hydrogen peroxide. This process was repeated three times to partially remove organic matter and induce discoloration, making it easier to visualize MP using a stereoscope. The observed fibers were identified as synthetic polymers as non-synthetic polymers, like cellulose, cotton, and silk, usually degrade during organic matter removal.

Lines 256-257 – What were the methodological differences?

We modified the paragraph as follows “The notable disparities observed between the two studies, conducted on the same coast with only a one-year gap, may be attributed to methodological variations or highlight a high temporal variability in MP abundance associated with Sargassum. [19] measured MP attached to fresh Sargassum but omitted those present in the rinsing liquid. Our study demonstrated that the latter can constitute up to 86% of the sample.”

A discussion of the limitations to the selected visual methodology is missing but critical to include. Fragments and dark colors are most definitely easier to spot visually than fragments or films and tr

A sentence has been added regarding the limitations of microplastic fiber identification. Colored fibers are easy to locate in the optical microscope, but the same happens with transparent fibers because they look very bright in this type of microscope. So, underestimating transparent fibers is not as likely as with transparent particles.

Additionally, the authors rebuttal notes that MP sizes were not recorded, but a size range would be a helpful characterization to understand the potential for ecological impact. There are size scales on the photos of particles. Were all particles photographed? Could the photos be analyzed to capture size information.

In our work, we counted 1042 microplastic fibers, of which 84 were precisely measured by electron microscopy. The data of these measurements, range, mean, standard deviation and a size-frequency distribution have been included in the methodology section and in the results.